# Family Needs Assessment of Patients with Cerebral Palsy Attending Two Hospitals in Accra, Ghana

**DOI:** 10.3390/children10081313

**Published:** 2023-07-29

**Authors:** Abena K. Aduful, Faye Boamah-Mensah, Mame Yaa Nyarko, Margaret L. Neizer, Yvonne N. Brew, Lovia A. Williams, Benedict N. L. Calys-Tagoe, Henry K. M. Ackun, Edem M. A. Tette

**Affiliations:** 1Department of Family Medicine, Korle-Bu Teaching Hospital, Accra P.O. Box GP 4236, Ghana; 2Princess Marie Louise Children’s Hospital, Accra P.O. Box GP 122, Ghananeizermargaret@gmail.com (M.L.N.); 3Department of Child Health, Greater Accra Regional Hospital (GARH), Accra P.O. Box GP 473, Ghana; 4Department of Child Health, Korle-Bu Teaching Hospital, Accra P.O. Box GP 4236, Ghana; 5Department of Community Health, University of Ghana Medical School, Accra P.O. Box GP 4236, Ghana; 6Department of Family Medicine, Duffus Health Center, Halifax, NS B3M 3Y7, Canada

**Keywords:** cerebral palsy (CP), family needs, disability, children, developmental disorder, healthcare, family-centered care, caregivers, accessibility, assistive devices

## Abstract

Background: The family represents the most essential and supportive environment for children with cerebral palsy (CP). To improve children’s outcomes, it is crucial to consider the needs of families in order to offer family-centered care, which tailors services to these needs. Objective: We conducted a needs assessment to identify the family needs of patients with CP attending two hospitals in Accra. Methods: The study was a cross-sectional study involving primary caregivers of children with CP attending neurodevelopmental clinics. Structured questionnaires were used to collect data spanning an 8-month period. The data were summarized, and statistical inference was made. Results: Service needs identified were childcare, counseling, support groups, financial assistance, and recreational facilities. Information needs included adult education, job training/employment opportunities, education, health and social programs, knowledge about child development, and management of behavioral and feeding/nutrition problems. Reducing extensive travel time was desirable to improve access to healthcare. With the increasing severity of symptoms came the need for improved accessibility in the home to reduce the child’s hardship, as well as assistive devices, recreational facilities, and respite for the caregiver(s). Conclusion: Families of children with CP have information, service, and access needs related to their disease severity and family context.

## 1. Introduction

Cerebral palsy is a permanent disorder of movement and posture development caused by non-progressive injury to the developing fetal/infant brain [1,2]. It is a major cause of chronic childhood disability worldwide and manifests in different ways which include delayed motor development, musculoskeletal problems, and comorbidities such as epilepsy, visual impairment, deafness, and speech impairment [3,4,5]. While studies have identified birth asphyxia, neonatal infections, kernicterus, and prematurity/low birth weight as the major cause of cerebral palsy in Africa [3], most studies in the United States and Europe have identified low birth weight, particularly prematurity, as the major risk factor [3,6].

Generally, the prevalence of cerebral palsy is 2.4 per 1000 children, but it varies from country to country, with the prevalence being higher in African countries than in Western countries [7,8]. Cerebral Palsy Africa estimates that 1 child per 300 births has cerebral palsy in Ghana [9].

Children with CP require ongoing services to adulthood [3,10]. Thus, to maximize their educational and developmental potential, children with cerebral palsy require support and multidisciplinary care [11]. Family-centered care is an approach to service delivery in which families work together with service providers in a way that considers their role in the child’s life to improve outcomes. This enables their expertise in knowing the child’s abilities and needs to be employed. Almasri et al. define family needs as a family member’s expressed desire for information, services, and support to achieve goals related to their family [12]. Family-centered care is therefore a set of attitudes, values, and approaches to service delivery for children with special needs and their families where the service is holistic and tuned to the preferences of the family in question while considering the contextual factors [13,14,15].

Research has found that family needs of children with cerebral palsy revolve around three main areas which are information needs, service needs, and access to healthcare, as shown in the conceptual framework (Figure 1). There is evidence that parental physical and mental health are often challenged when handling a child with cerebral palsy, and this can, in turn, affect the child [16,17,18,19,20]. Although this is the best practice in rehabilitation care [15,21,22,23,24,25], the evidence shows that healthcare in many places is still more child-centered [21,23,26,27]. This is because health professionals have difficulties in determining their role in identifying family needs and making necessary arrangements to meet these needs, leading to missed opportunities and parental dissatisfaction [23,28,29].

Since it is recommended that service providers explore the individualized family needs of children with disabilities to offer needs-based family-centered care [12], this study set out to determine the various needs of families of children with cerebral palsy to design a model for providing services tailored to meet these needs.

## 2. Materials and Methods

### 2.1. Study Design

The study was a needs assessment that employed a cross-sectional design to explore the service, information, and access to healthcare needs of families with children who have cerebral palsy. The study was conducted at the Neurodevelopmental Clinics at Princess Marie Louise Children’s Hospital and Greater Accra Regional Hospital both in Accra, Ghana, from June 2022 to January 2023.

### 2.2. Study Site

Princess Marie Louise Children’s Hospital (PMLCH) is located in the business capital of the Accra Metropolis, in the Ashiedu Keteke sub-metropolitan area. It offers both primary and specialist care to children between the ages of 0 and 18 years; thus, it receives both self-referred patients and referrals from health facilities in Accra and the rest of the country. It is the largest children’s hospital in the country and has a bed capacity of 110 beds with a total clinical staff of 168, including 4 resident pediatricians and 113 non-clinical staff. Daily Outpatient Department (OPD) and Emergency Room (ER) attendance are at an average of 152 patients daily with an average of 2168 admissions a month.

The Greater Accra Regional Hospital (GARH) is situated within the Korle Klottey Metropolitan Assembly. The hospital is a referral and training center that provides 24-hour general and specialist health services and has a bed capacity of 420 beds, including 100 pediatric beds. It has all the various departments in a tertiary facility except a few specialist areas such as a cardiothoracic center and genetics. It has 9 specialists and an average OPD attendance of 1292 a month, and an emergency room attendance of 186 a month. The neurodevelopmental clinic attends to an average of 27 patients a month. 

### 2.3. Study Population

The target population for this study was the caregivers of children with cerebral palsy attending neurodevelopment clinics at PMLCH and GARH.

### 2.4. Inclusion and Exclusion Criteria

All caregivers of pediatric patients aged 1–17 years with cerebral palsy who attended the neurodevelopmental clinic at PMLCH and GARH in the past two years were eligible for the study. Caregivers who consented to be part of the study were included in the study. They were excluded if they were not the primary caregivers of the child with cerebral palsy, if their child’s diagnosis was unclear or if the case notes could not be found, if the child was acutely ill or unwell during the period of data collection, if the diagnosis was also linked to another major neurodevelopmental condition such as a known metabolic disease, and if the primary caregivers themselves were unwell or were unable to communicate.

### 2.5. Sampling Method and Sample Size Determination

The study adopted a census survey [30] involving the targeting of all patients with cerebral palsy and their respective caregivers who attended the neurodevelopmental clinics at both study sites 2 years prior to the start of data collection. There were 76 and 73 registered cerebral palsy patients at PMLCH and GARH, respectively, within the two years from 2020 to 2021, giving a total of 149 patients who were invited to participate in the study. Other studies on cerebral palsy patients in Ghana recruited 75 and 76 cerebral palsy patients/caregivers, respectively [31,32].

### 2.6. Data Collection Techniques

The data were collected over an 8-month period using a structured questionnaire that was administered to caregivers by the researchers from June 2022 to January 2023. The interviewer-administered structured questionnaires were administered mostly through face-to-face interviews. Phone interviews were added as an optional means of data collection. The informants were primary caregivers where the primary caregiver was defined as the person who spends the most time with the child, performs most of the caregiving tasks, and has the duty of taking care of the child [33,34]. Data collection was carried out by health professionals in these hospitals trained by the investigators. Prior to administering the questionnaires, informed consent was obtained from participants. It was also pointed out that if they needed any further psychological support, it could be offered at no cost to them at the neurodevelopmental clinic and on the phone.

### 2.7. Data Collection Tools

#### 2.7.1. The Questionnaire

The questionnaire for the study contained questions on the socio-demographic characteristics of the families and the child’s Gross Motor Function Classification Score (GMFCS). The rest of the questionnaire was largely adopted from *Building A Full-Service School: A Step-by-Step Guide* by Carol Calfee, Frank Wittwer, and Mimi Meredith [35]. The questionnaire was pretested, and the necessary modifications were made to the questions constructed by the authors that had been added to the family needs assessment tool by Calfee et al.

#### 2.7.2. Family Needs Assessment Tool

The family needs assessment tool created by Carol Calfee, Frank Witter, and Mimi Meredith was used to measure family needs in this study. The questions in the tool were grouped into three (3) groups of needs: information needs, service needs, and access to healthcare needs. The tool has seven questions that assess information needs which include questions about the child’s development, the child’s behavior problems, programs that can help the child, adult education opportunities, help with a specific problem, nutrition, and job training and employment opportunities. The seventeen questions that assessed service needs included questions about counseling, financial help, daycare, improving accessibility in the home, special equipment/assistive devices, and networking. Only one question assessed access to healthcare needs, which was a question on better healthcare for the child. Five additional questions were added by the researchers that assessed access to healthcare needs with questions on healthcare costs, long waiting times, extensive travel time or distance from the hospital, convenience of appointment times, and doctors/nurses rushing through clinics, together with one other question that assessed service needs with respect to the need for a recreational facility for the child.

#### 2.7.3. The Gross Motor Function Classification System—Expanded & Revised (GMFCS-E&R)

The GMFCS E&R is a scoring system that leads to a five-level classification that differentiates children with cerebral palsy based on their current motor function and needs for assistive technology and wheeled mobility [36]. Using this system, a child can have a score of 1 to 5, which is their Gross Motor Function Classification Score (GMFCS). A score of 1 indicates a level of advanced development of motor skills or independence in mobility, and a score of 4 or 5 indicates lower competence in motor skills or dependence on assistive devices. Thus, children with lower scores are more likely to be independently mobile while those with higher scores are more likely to require assistive devices for mobility. It uses features of the child such as neck control and the ability to sit and walk [37]. The GMFCS level was used as the independent variable.

### 2.8. Data Handling and Analysis

Researchers checked the completed questionnaires for omissions, incomplete answers, or unclear statements after administration in the presence of the caregivers, and any errors or omissions were corrected before the caregivers left. The data were entered into a Microsoft Excel 365 database after which data cleaning and sorting were carried out. Descriptive analysis was performed using Microsoft Excel and presented in a frequency table and figures. Means and standard deviations were also computed. Inferential statistics were applied (Spearman’s rank correlation) using the Statistical Package for the Social Sciences (SPSS) version 27. The GMFCS, which was an independent variable in this study, was compared with the dependent variables, which were the various needs, to determine whether there was any correlation between them. 

### 2.9. Ethical Consideration

Ethical approval was sought and obtained from the Ghana Health Service Ethics Review Committee (GHS-ERC: 008/01/22). The publishers of the book *Building A Full-Service School: A Step-by-Step Guide* by Carol Calfee, Frank Wittwer, and Mimi Meredith were contacted and gave permission for the team to use the family needs assessment tool in their book (license no. 5117260346828). Confidentiality was ensured. Consent was sought from all study participants.

## 3. Results

### 3.1. Sociodemographic Characteristics

The 149 children with cerebral palsy and their caregivers who attended the two neurodevelopmental clinics over the past two years, and were the target of this census survey, were invited to take part in the study. However, this number was not reached as 20 caregivers declined and 29 others were lost to follow-up and not available. Thus, an additional 20 patients who registered at the clinic in 2022 were included, resulting in a total of 120 participants.

Table 1 provides a summary of the sociodemographic characteristics of the study population. Altogether, 81% of patients were aged 1–5 years, with the minority aged 11–17 years (2%, *n* = 3). The ages ranged between 12 months (1 year) and 156 months (13 years), with a mean age of 43 months (SD = 32.3). Most patients were first seen at the neurodevelopmental clinic between the ages of 0 months and 12 months (53%, *n* = 64). The mean age at first presentation was 25 months (SD = 28.8). It ranged from 1 week to 132 months. Almost two-thirds of the patients were male (63%, *n* = 76). The most common ethnic group was Akan (43%, *n* = 52). There was an equal proportion of patients, 30% (*n* = 36), with GMFCS of 4 and 5. The majority of children of school-going age above 2 years of age had never been to school (65%, *n* = 55), and 6% (*n* = 5) of them had left school, with only 29% (*n* = 24) being enrolled in school. More than half (56%, *n* = 67) of the households had only two adults in the family and 75% (*n* = 90) of the children had one or more siblings at home. The majority, 77%, of the parents were married.

A significant number of caregivers were the mothers of the children (82%, *n* = 98), with fathers being the sole caregivers in only 2% (*n* = 3) of cases. In all, 8% (*n* = 10) of the caregivers were solely grandparents. More than half (58%, *n* = 69) of the caregivers had not attained senior high school certification. The majority of families earned less than 1000 Ghana Cedis (about USD 100) a month (57%, *n* = 69). Almost all the patients had an active National Health Insurance Scheme (NHIS) card (96%, *n* = 115). Of the remaining 4% (*n* = 5), 3% depended on private health insurance. A significant number of families had no social welfare support (97%, *n* = 116).

### 3.2. Service Needs

Figure 2 provides a summary of the service needs reported by caregivers. In all, 88% (*n* = 105) of caregivers really needed more money or financial help, and more than half of the caregivers, 53% (*n* = 63), really needed help from someone who can babysit for the day or evening so they could get away. The majority of caregivers really needed help with finding recreational facilities for their children 76% (*n* = 91), assistive devices to meet their child’s needs 81% (*n* = 97), improved accessibility in-home to reduce the child’s discomfort 62% (*n* = 74), and counseling to cope with their situation 73% (*n* = 87). Only about a quarter of the caregivers, 26% (*n* = 31), reported that they really needed help to deal with problems with in-laws or other relatives. In all, 63% (*n* = 76) did not need help with this. While the majority, 74% (*n* = 89) did not need help with problems concerning their spouses, 17% (*n* = 20) really needed help and 9% (*n* = 11) needed some help. More than half, 54% (*n* = 65), really needed help with more friends who have a child like theirs.

### 3.3. Information Needs

Figure 3 provides a summary of the information needs reported by the caregivers. The majority of caregivers, 85% (*n* = 102), really needed help with more information about their child’s development. Similarly, most caregivers, 79% (*n* = 95), really needed more information about behavioral problems. Almost all respondents, 92% (*n* = 110), really needed more information about programs that can help their children (education, health, social). M ost caregivers, 63% (*n* = 75), really needed help with more information about nutrition or feeding, while 43% (*n* = 51) of the caregivers really needed help with information about adult education opportunities. Additionally, most caregivers, 63% (*n* = 51), really needed help with information about job training and employment opportunities for themselves.

### 3.4. Access to Healthcare Needs

Figure 4 provides a summary of caregivers’ responses to obstacles hindering access to healthcare. More than half of the caregivers (58%, *n* = 69) did not need help with waiting times to see the doctor. Almost half of the caregivers (48%, *n* = 58) really needed help with extensive travel time or distance from the hospital, a fifth (20%, *n* = 24) needed some help, and about a third did not need help (32%, *n* = 38). Only 11% (*n* = 13) reported that they really needed help with appointments not being available at convenient times. Similarly, only 8% (*n* = 10) of caregivers reported that they really needed help with doctors/nurses rushing through clinic visits, but the majority, 80% (*n* = 96), did not need help with this.

### 3.5. Correlation between GMFCS and Family Needs

There was a positive correlation between the needs displayed in Table 2 below and the GMFCS. All other needs showed no significant correlation with the GMFCS.

## 4. Discussion

In this study, 81% of the patients were aged 1–5 years, with a minute minority (2%) aged 11–17 years. Research on cerebral palsy in Ghana found that over half of the caregivers who started medical treatment eventually transitioned to full home care [38]. The reasons for this included transportation costs, difficulty traveling long distances, long waiting times, negative experiences with healthcare workers, and the belief that they can manage children effectively at home [38]. These issues may have accounted for the small proportion of older children seen in this study. There have been many reports that the male sex carries a higher risk of cerebral palsy [39]; the finding that two-thirds of the children with CP in this study were males further strengthens this assertion. Akans were the most common ethnic group (43%) since they are the largest ethnic group in Ghana, forming 47.5% of the population [40]. Gas and Ewes were also dominant because the data were collected in the Greater Accra Region where the Gas live and Accra is multiethnic.

More than half (53%) of the patients had been referred to the neurodevelopmental specialist between the ages of 7 days and 12 months, indicating that they were referred early to the neurodevelopmental clinics. This is a positive finding since children with cerebral palsy have a critical window of neuroplasticity and referring them early for early intervention is generally associated with better outcomes [41]. More than half (60%) of the children in our study had GMFCS of 4 and 5 cumulatively. Severely disabled children who have GMFCS of 4 and 5 require additional healthcare resources, have more family needs, and have higher medication costs, as they present with more medical problems compared to those with lower scores [36,42].

Some parents identified services as their greatest need. Our findings showed that 81% of families needed assistive devices/special equipment, 53% needed childcare, 76% needed recreational facilities for their child, and 62% needed improved accessibility in the home to reduce the child’s discomfort. We also found that all these needs positively correlated with the GMFCS or symptom severity. Families have reported common needs such as occupational therapy, speech and language therapy, physical therapy, and assistive devices, and that their homes are not handicap-accessible as the child grows and they find it difficult to lift or carry the child around. Others cite service needs such as recreational/entertainment activities and childcare for both recreation and emergencies [43,44]. Nursery or school placement is one of the ways to meet these needs as it provides respite and recreational opportunities. However, we found that a significant number of these children have never been to school. Children with cerebral palsy faced challenges in obtaining educational facilities because the schools do not have the facilities or personnel to cater to them or the capacity to support them in a special school setting [45]. Those who start mainstream school drop out for these same reasons. This could explain why three out of five children of school-going age in our study had never been to school; thus, the government must create special nurseries and schools, as well as places for respite care.

Receiving psychological help in dealing with the child’s condition is also a need [42]. Close to three-quarters of caregivers in this study expressed the need for psychological assistance to deal with their child’s condition, and these parents needed to be referred for the necessary help. Parents of children with cerebral palsy have indicated that their child’s diagnosis affects them mentally and socially [9]. Their stress levels are associated with the severity of the child’s disability [46]. A family needs assessment in Northern Nigeria also found that mothers have a need to discuss their feelings such as stress and depression with someone with a similar experience [47]. This further emphasizes the importance of support groups and access to psychological support in these clinics [48]. 

In this study, three out of four of the parents were married. This is a positive finding since studies have found that spouses who had children with cerebral palsy were the most supportive of each other [49,50]. Furthermore, family relationships that have more cohesion, expressiveness, and less conflict have been found to be associated with fewer needs when caring for a child with a disability [51]. Our study showed that most families (74%) did not need help with spousal problems. Caregivers, mostly mothers, also indicated that they did not need help with issues concerning their in-laws (63%). This is in contrast with other studies in Ghana, which found that there was tension in the nuclear family as well as the extended family, stemming from the diagnosis of a child with cerebral palsy [52]. Researchers in China also found that mothers of children with cerebral palsy face family conflicts and rejection from parents-in-law because of public shaming and criticism [53]. Our findings with respect to this were positive and commendable. 

A significant proportion (88%) of families in this study reported that they really needed financial help, with only 7% not needing it. The majority (57%) of families earned less than GHC 1000 a month (i.e., about USD 2.8 a day) for the whole family, implying that an individual in these families is likely to spend less than USD 2.15 a day, putting them in the extreme poverty range of the World Bank [54]. Raising a child with cerebral palsy comes with some amount of financial strain [55]. Most caregivers need money for medications not covered by insurance, vehicles with a wheelchair lift, adaptive equipment, more special therapies, and basic necessities such as diapers. They also face the dilemma of how to make sure their child is financially cared for once they pass away [43,47,56]. Additionally, these families have been noted to earn less [57], usually because one member of the family—in most cases, the mother—has to stay at home and cater to the child [45]. Altogether, 82% of the primary caregivers in our study were mothers. Although social welfare support systems and other government support systems are available, most caregivers are either not aware of them or do not have access to them, as previously reported [58]; thus, we found that an overwhelming number (97%) did not have any social welfare support. Additional support systems and creating awareness of the existing support systems are therefore required, as well as opportunities for caregivers to earn an income. The National Health Insurance Scheme (NHIS) has been put in place to make healthcare more accessible to families with additional needs, though not all medical conditions are covered by the scheme [59,60]. Finding that 96% of the children were on the NHIS was a positive finding, although fathers in earlier studies have complained that they usually had to cover the cost of medications, as these medications were not covered by the scheme [59,60].

The need for information still ranks as an unmet need for caregivers of children with cerebral palsy [42,43,47,61]. Moreover, since these children are special, they require a high level of physical and emotional support which comes from training and the impartation of knowledge to caregivers, which enables them to understand their children’s diagnosis and articulate and handle their challenges. We found that the majority (58%) of the caregivers had not attained senior high school certification; thus, caregivers in this setting may be limited in the amount of information they have about rehabilitation, training, and feeding [9,31,48], unless the education process is carefully planned. In research comparing Scotland and Hong Kong concerning parental responses to healthcare services for children with chronic conditions, effective communication with parents was recognized as a key to effective medical practice in both countries [62]. Information related to diagnosis and treatment was also the best predictor of parental rating of the quality of a clinic.

While this study showed that parents needed knowledge on managing their children and mentoring them, other caregivers of children with CP in Ghana have reported having limited information about their child’s diagnosis, rehabilitation, training, and feeding [9,31,32,48]. Rawling et al. have summarized the information needs as follows: (1) understanding how the disease/condition affects the child’s growth and development, (2) identifying community resources, (3) provision for the child’s special education/intellectual needs, (4) behavior management/discipline, and (5) improvement of communication among their child’s healthcare providers [61]. Our study discovered that information was really needed in all these areas. Thus, the education of parents in these clinics should be a priority, carefully planned, and reflect the language of the majority ethnic groups.

Long distance/extensive travel time to clinics, long waiting times, relatively high healthcare costs, inconvenient appointment times, and fragmentation of care have been identified as primary obstacles to accessing healthcare [61,63,64]. In resource-limited settings, diagnostic facilities such as Magnetic Resonance Imaging (MRI) and professionals with expertise in managing cerebral palsy are often found in big cities and lacking in rural areas [3,65]. Furthermore, modern interventions such as botulinum toxin and Vojta therapy are very limited [66]. Almost half (48%) of caregivers really needed help with extensive travel time/distance from the hospitals in our study, and a further 20% needed some help with this. Similarly, a study involving 40 primary caregivers found that all of them expressed concern about the inaccessibility of specialist clinics where they lived; hence, they had to travel long distances to access healthcare in the Greater Accra Region [9]. Therefore, advocacy is required for a government policy to support the transportation costs of caregivers and to increase the number and distribution of multidisciplinary clinics providing assessment and therapy services comparable to a “one–stop-shop”. 

Opportunities exist for healthcare providers to engage families for convenient appointment times and reduce waiting times. Our findings revealed that most of the caregivers did not need help with this. Likewise, the majority (80%) felt that nurses and doctors did not rush them through the clinics but took their time with them. These were positive findings. In contrast, a similar study carried out in Indiana found that although parents tend to understand the pressure on nurses and doctors, they stated that it would be much more helpful and appreciated if they are not hurried or rushed through clinic visits [43]. 

The main limitation of this study was its location in the capital city, which mainly provided an urban perspective of the needs in one of the sixteen regions of the country. Additionally, being a facility-based study, it may not reflect the peculiar needs of families whose children with CP are homebound and have limited access to services.

## 5. Conclusions

Children with cerebral palsy in Ghana have needs with respect to services, information, and access to healthcare. Some of these needs are related to the child’s GMFCS. Quantifying these specific needs has provided an opportunity to offer family-centered care tailored to meet the felt needs of patients and their caregivers. Capitalizing on this will help achieve optimum and inclusive healthcare for children with cerebral palsy in Ghana.

## Figures and Tables

**Figure 1 children-10-01313-f001:**
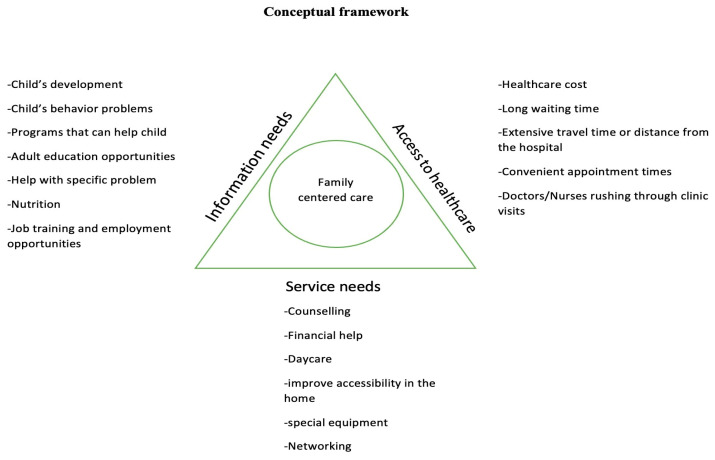
Model for assessing family needs in family-centered care.

**Figure 2 children-10-01313-f002:**
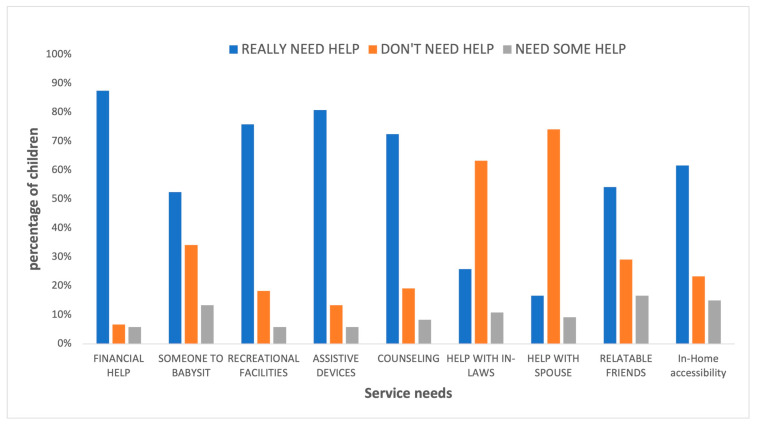
Service needs of the families of children with cerebral palsy reported by caregivers (*n* = 120).

**Figure 3 children-10-01313-f003:**
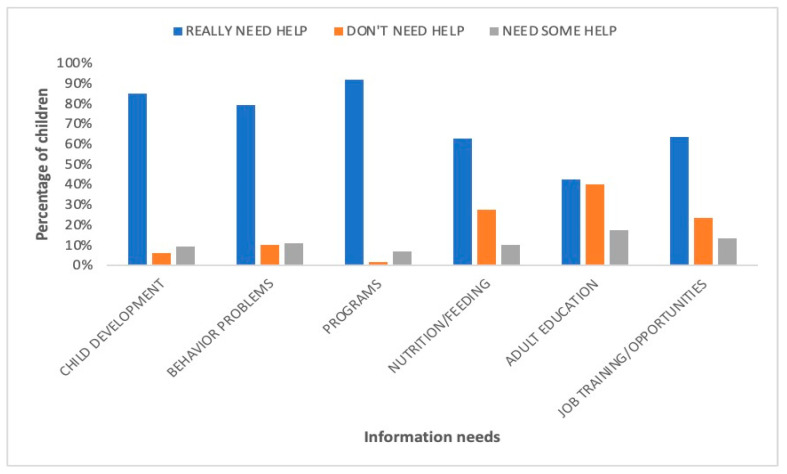
Information needs of the families of children with cerebral palsy reported by caregivers (*n* = 120).

**Figure 4 children-10-01313-f004:**
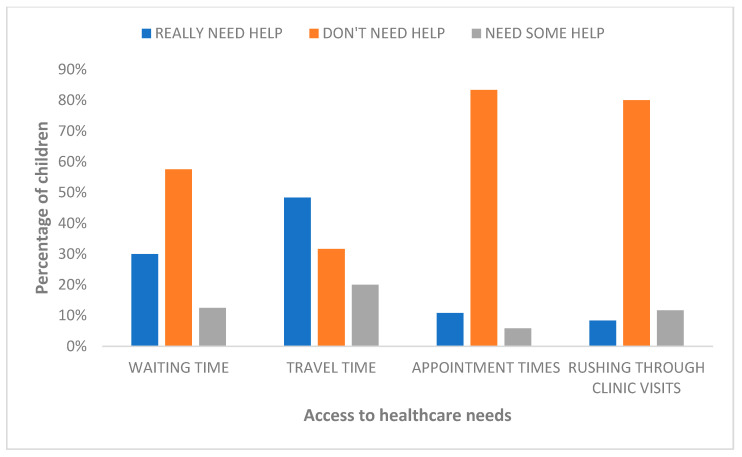
Obstacles hindering access to healthcare reported by caregivers of children with cerebral palsy (*n* = 120).

**Table 1 children-10-01313-t001:** Sociodemographic characteristics of the participants.

Characteristics of Participants	Number (*n*)	Percentage (%)
**Characteristics of the Children**		
Gender		
Male	76	63
Female	44	37
Age (months)		
12	12	10
13–24	36	30
25–60	49	41
61–132	20	17
133–204	3	2
Age Seen at Clinic (months)		
0–12	64	53
13–24	28	23
25–60	19	16
61–132	7	6
133–204	2	2
GMFCS		
I	18	15
II	16	13
III	14	12
IV	36	30
V	36	30
Insurance		
NHIS *	115	96
Private Health Insurance	4	3
Expired NHIS	1	1
Social Welfare Assistance		
Yes	3	3
No	117	97
**Characteristics of Caregivers**		
Marital Status		
Married	92	77
Separated	6	5
Never married	16	13
Divorced	4	3
Widowed	2	2
Monthly Family Income (GHC)		
Less than 1000	69	57
1001–2000	36	30
2001–3000	12	10
3001–4000	2	2
More than 4000	1	1
Ethnicity#		
Ga	25	21
Guan	1	1
Ewe	25	21
Akan	52	43
Hausa	8	7
Other	9	7
Primary Caregiver’s Relationship with Child		
Mother	98	82
Grandparent	10	8
Mother and Father ^†^	3	3
Mother and Grandparent ^†^	4	3
Father	3	2
Mother and Father and Grandparent ^†^	1	1
Auntie	1	1
Primary Caregiver Completed SHS		
Yes	51	42
No	69	58

Note. *n* = 120; * NHIS = National Health Insurance Scheme; ^†^ shared caregiving

**Table 2 children-10-01313-t002:** Spearman’s rank correlation test showing positive correlations between the needs of caregivers and the child’s GMFCS.

Needs of Caregivers	*p*-Value	r(118)
Improved accessibility in the home to reduce hardship for child	<0.001	0.3
Assistive devices to meet my child’s needs	0.003	0.27
More time for myself	0.017	0.24
Recreational facilities for my child	0.009	0.22

## Data Availability

Data used in this study belong to Princess Marie Louise Children’s Hospital of the Ghana Health Service. The data are available upon request to researchers who meet the criteria for access to confidential data. Requests for the data should be directed to Dr. Abena K. Aduful at adufulabena32@gmail.com. Any request for the data should provide details of what the data are supposed to be used for and should meet the requirements of the Ethical Review Committee of the Ghana Health Service and be used for the purpose defined in the request or research protocol.

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
