# Peer review of "Family Needs Assessment of Patients with Cerebral Palsy Attending Two Hospitals in Accra, Ghana"

_children, 2023, doi:10.3390/children10081313_

Round 1

Reviewer 1 Report

Thank you for the pleasure and opportunity for me to review your paper about the family needs of patients with cerebral palsy in Ghana. I found it an interesting read. I would like to preface that as an orthopaedic doctor I am not that familiar with qualitative studies, and what the gauge is for quality studies in this context. 

Abstract

- clear and effective 

- nil issues 

Introduction 

- very well written

- good background information

- It however quite long and would recommend cutting down.

- additionally using graphs/figures are not normal practice and wonder if it is necessary. 

methods 

- the methodology is clear however I wonder if it can be substantially reduced in wording 

- I wonder however if this truly is a "quantitative" study. All outcomes use qualitative data and only the correlation with the GMFCS utilises statistics. The rest is descriptive statistics of pts qualitative outcomes.

- Reference for each questionnaire should be attached - GMFCS, Building, Family needs etc. Line 154, 164

Results

- well displayed and clear 

Discussion 

- once again well written 

- however here i do believe the discussion is definitely too long. There is alot of information and it rather feels like a literature review than a disucssion.

- The discussion would greatly benefit with how the results of this study translates to specific ways of changing practice and for better patient care

- once again i disagree with this being a quantitative study in line 481. Pts answering "really need help" "dont need help" "need some help" does not apply to quantitative data. If there was an explicit, validated scoring system it would be a different case.

nil further

Author Response

Department of Community Health

University of Ghana Medical School

  1. O. Box 4236

                                            Accra

                          19th July, 2023

Dear Sir/Madam,

Response to reviewer’s comments and submission of revised Manuscript ID number: children-2483170

Title of paper: Family needs assessment of patients with cerebral palsy attending two hospitals in   Accra, Ghana.

Thank you for your positive comments about this manuscript and recommended changes to improve it. Our response to the comments are as follows:

REVIEWER 1

Comments and Suggestions for Authors

Thank you for the pleasure and opportunity for me to review your paper about the family needs of patients with cerebral palsy in Ghana. I found it an interesting read. I would like to preface that as an orthopaedic doctor I am not that familiar with qualitative studies, and what the gauge is for quality studies in this context. 

  1. Introduction 

i)- very well written, good background information. It however quite long and would recommend cutting down.

The introduction has been modified and reduced.

  1. ii) - additionally using graphs/figures are not normal practice and wonder if it is necessary. 

We used the diagram to illustrate the conceptual framework behind this study as it indicates the relationships between the three branches of family needs and how they are related to family centered care. Thus, it provides a succinct summary of the main idea behind this study which will be helpful for readers particularly those who are not in this field and therefore we would like to retain it.

  1. Methods 

(i)- the methodology is clear however I wonder if it can be substantially reduced in wording 

We have reduced the methodology section

(ii)- I wonder however if this truly is a "quantitative" study. All outcomes use qualitative data and only the correlation with the GMFCS utilises statistics. The rest is descriptive statistics of pts qualitative outcomes.

We admit that we have used descriptive statistics in this study and most of our variables are categorical so it resembles a qualitative study as you suggested. However, the questionnaire had a quantitative design because we used the Likert scale with 3 options to choose from for each question. They were 1. “I really need some help in this area” 2.” I’d like some help but my need is not that great” 3. I don’t need any help in this area.” This is recognized as a tool for quantitative research [South, L., Saffo, D., Vitek, O., Dunne, C., & Borkin, M. A. (2022). Effective use of Likert scales in visualization evaluations: A systematic review. In Computer Graphics Forum (Vol. 41, No. 3, pp. 43-55)]. We have removed the label and left only the cross-sectional design to avoid confusion.

iii) Reference for each questionnaire should be attached – GMFCS ( ), Building, Family needs etc. ( ) Line 154, 164

Page 4 paragraph 2 line 4 and paragraph 4 lines 10 (Lines 140 and 165, 171): The References have been applied as:  

  • …’Building A Full-Service School: A Step-by-Step Guide’ by Carol Calfee, Frank Wittwer, and Mimi Meredith [35].

  • It uses features of the child such as their ability to have neck control, sit and walk [37]...”

3.Results -- well displayed and clear 

No comments

 Discussion 

- once again well written, however here I do believe the discussion is definitely too long. There is a lot of information and it rather feels like a literature review than a discussion.

The discussion has been modified and shortened.

- The discussion would greatly benefit with how the results of this study translates to specific ways of changing practice and for better patient care

Page 9 to 12: This has been done and the sentences have been highlighted in bold in the text.

- once again I disagree with this being a quantitative study in line 481. Pts answering "really need help" "dont need help" "need some help" does not apply to quantitative data. If there was an explicit, validated scoring system it would be a different case.

See comment in 2 (ii)

Minor editing of English language required

The English language has been edited

REVIEWER 2

Comments and Suggestions for Authors

Overall, the authors should be commended for their work and presentation of their research.

However, there are several small changes that should be performed before accepting this manuscript.

Yours sincerely

Prof  Edem Tette

Reviewer 2 Report

Overall, the authors should be commended for their work and presentation of their research.

However, there are several small changes that should be performed before accepting this manuscript.

Reviewers Comments

Page 4, Lines 136-140

The study adopted a census survey targeting all patients with cerebral palsy and their respective caregivers who attended the neurodevelopmental clinics at both study sites 2  years prior to the start of data collection. There were 76 and 73 registered cerebral palsy  patients at PMLCH and GARH respectively within the two years from 2020 to 2021 giving  a total of 149 patients who were invited to participate in the study.

Reviewer’s comments: Please clarify the meaning of ‘census survey’.

Page 4, Lines 158-160

The rest of the questionnaire was largely adopted from ‘Building A Full-Service School: A Step-by-Step Guide’ by Carol Calfee, Frank Wittwer, and Mimi Meredith [41]. The questionnaire was pretested, and the necessary modifications and clarification of terms were made before data collection.

Reviewer’s comments: Please clarify, was the questionnaire pretested and modified by Calfee et al? Were these pre-tests and modifications performed by the authors of this manuscript? Or, was the family needs assessment tool mentions in lines 164-165 used as the questionnaire?

Page 4, Lines 164-165

The family needs assessment tool created by Carol Calfee, Frank Witter, and Mimi Meredith was used to measure family needs in this study.

Reviewer’s comments: IF allowed by the authors (Carlfee et al.), it would be beneficial to add can the family needs assessment tool to the manuscript as supplemental data.

Page 5, Lines 188-189

It uses features of the child such as their ability to have neck control, sit and walk [43]. This  variable was used as the independent variable.

Reviewer’s comment: Please consider rewording the last sentence to “The GMFCS level was used as the independent variable.”

Page 5, Lines 217-219

Researchers checked the completed questionnaires for omissions, incomplete answers, or unclear statements after administration. After correcting the errors detected, the data was entered into an Excel Microsoft 365 database. Data cleaning and sorting were done.

Reviewer’s comment: Please clarify how these errors were corrected. Were the questions omitted, were the families contacted?

Author Response

Department of Community Health

University of Ghana Medical School

                                                                                                        P. O. Box 4236

                                            Accra

                          19th July, 2023

Dear Sir/Madam,

Response to reviewer’s comments and submission of revised Manuscript ID number: children-2483170

Title of paper: Family needs assessment of patients with cerebral palsy attending two hospitals in   Accra, Ghana.

Thank you for your positive comments about this manuscript and recommended changes to improve it. Our response to the comments are as follows:

REVIEWERS COMMENTS 2

Page 4, Lines 136-140

The study adopted a census survey targeting all patients with cerebral palsy and their respective caregivers who attended the neurodevelopmental clinics at both study sites 2 years prior to the start of data collection. There were 76 and 73 registered cerebral palsy  patients at PMLCH and GARH respectively within the two years from 2020 to 2021 giving  a total of 149 patients who were invited to participate in the study.

 Please clarify the meaning of ‘census survey’.

The census survey refers to taking the entire study population as the sample. It is used in studies where the population is small.( Israel, G. D. (1992). Determining sample size. https://www.psycholosphere.com/Determining%20sample%20size%20by%20Glen%20Israel.pdf)

Page 4, Lines 158-160

The rest of the questionnaire was largely adopted from ‘Building A Full-Service School: A Step-by-Step Guide’ by Carol Calfee, Frank Wittwer, and Mimi Meredith [41]. The questionnaire was pretested, and the necessary modifications and clarification of terms were made before data collection.

Reviewer’s comments: Please clarify, was the questionnaire pretested and modified by Calfee et al? Were these pre-tests and modifications performed by the authors of this manuscript? Or, was the family needs assessment tool mentioned in lines 164-165 used as the questionnaire?

Page 4 paragraph 2 lines 5-7 or the last 3 lines (Line 141 to 143): This has been clarified in the text to read: “The questionnaire was pretested, and the necessary modifications were made to the questions constructed by the authors that had been added to the family needs assessment tool by Calfee et al.” Questions from the family needs assessment tool from Calfee et al was not modified.

Page 4, Lines 164-165

The family needs assessment tool created by Carol Calfee, Frank Witter, and Mimi Meredith was used to measure family needs in this study.

Reviewer’s comments: IF allowed by the authors (Carlfee et al.), it would be beneficial to add can the family needs assessment tool to the manuscript as supplemental data.

Permission has not been sought to publish tool as part of this research. However, reviewers can view tool at this link;  https://www.pdffiller.com/jsfiller-mob11/?mode=force_choice&requestHash=e00d002140671ec3af5f85fa6455fcedfd0e09c0a91fd1324b9bf73c8538e4d5&projectId=1305556395&jsf-page-rearrange-v2=false&jsf-new-header=false#4d45e6480df0460587564bcd95b7fc9d

Page 5, Lines 188-189

It uses features of the child such as their ability to have neck control, sit and walk [43]. This variable was used as the independent variable.

Reviewer’s comment: Please consider rewording the last sentence to “The GMFCS level was used as the independent variable.”

Page 4, Paragraph 4 lines 9-11 or last two lines (Lines 184-186): We have modified the sentence to read: “It uses features of the child such as their ability to have neck control, sit and walk [37]. The GMFCS level was used as the independent variable.”

Page 5, Lines 217-219

Researchers checked the completed questionnaires for omissions, incomplete answers, or unclear statements after administration. After correcting the errors detected, the data was entered into an Excel Microsoft 365 database. Data cleaning and sorting were done.

 Reviewer’s comment: Please clarify how these errors were corrected. Were the questions omitted, were the families contacted?

Page 4 paragraph 5 lines 1-3:  We have explained how this was done by modifying the statement to read “Researchers checked the completed questionnaires for omissions, incomplete answers, or unclear statements after administration in the presence of the caregivers, and any errors or omissions were corrected before the caregivers left.” The data were entered into a Microsoft Excel 365 database after which data cleaning and sorting were done.

Yours sincerely

Prof  Edem Tette
